# DISTILL MODELS BY APTITUDE: EFFICIENT REASONING CAPABILITY DISTILLATION VIA ADAPTIVE DATA CURATION AND OVERTHINKING MITIGATION

## ABSTRACT

The exponentially increasing computational demands of large language models (LLMs) facilitate the distillation of knowledge or capability to smaller models. Existing distillation attempts to transfer LLMs' reasoning capabilities to compact models face critical limitations, including expensive training or annotation costs, suboptimal data selection, and flawed synthetic data due to LLMs' general tendency to overthink. This paper introduces DynaGuide, a novel framework that optimizes the distillation process in both efficiency and performance. Our approach integrates (1) Dynamic Data Selection that adaptively performs fine-grained valuable data selection during the training process, and (2) Reasoning Pattern Guidance that mitigates the overthinking problem in synthetic data by incorporating specialized guidance during fine-tuning. Extensive experiments demonstrate that DynaGuide consistently achieves stable performance improvements across models of different series and parameter scales, with gains surpassing those of baseline methods on knowledge reasoning question answering benchmarks. Our systematic ablation studies and analysis further provide valuable insights into distillation and reasoning.

## 1 INTRODUCTION

The rapid evolution of AI has witnessed a dramatic surge in model complexity, progressing from early small models to today's large language models (LLMs) that exhibit remarkable reasoning capabilities. However, this advancement comes at an exponential increase in training costs, creating significant computational and financial barriers (Cottier et al., 2024). Although knowledge distillation is thought as a promising solution to this challenge by transferring LLMs' excellent reasoning capabilities to more compact and efficient models using LLM-generated synthetic data (Xu et al., 2024b), distillation based on large datasets (Guo et al., 2025; Yu et al., 2025) remains computationally intensive and time-consuming. Alternative approaches explore to use only a small amount of data for distillation, but introduce costly human experts annotations (Ye et al., 2025), or adopt coarse-grained data selection and ignore the adaptability to the model (Team, 2025; Muennighoff et al., 2025).

Moreover, recent studies have found that reasoning LLMs generally suffer from overthinking (Chen et al., 2024). Such models can get the correct answer at early reasoning stages (Fu et al., 2024), but continue the thinking process with much verification of previous steps or exploration of other unnecessary reasoning paths (Chen et al., 2025), generating redundant thinking tokens and reducing inference efficiency (Sui et al., 2025). Even worse, frequent verification and transition can disrupt reasoning continuity, degrade contextual coherence, reduce reasoning depth, and ultimately result in lower performance (Wang et al., 2025). When such flawed synthetic data is used for distillation, it can be more difficult for small models to acquire robust knowledge reasoning ability, thus more challenging to maintain efficiency and accuracy.

To address these limitations, we propose **DynaGuide**, a novel distillation framework that efficiently transfers the knowledge reasoning capability of LLMs to small models. As shown in Figure 1, DynaGuide includes two key components: **Dynamic Data Selection (DDS)** and **Reasoning Pattern Guidance (RPG)**. DDS performs adaptive data selection during the training process, similar to the idea of active learning (Cohn et al., 1996), where a small number of the most valuable samples are selected for training. Differently, we have access to the metadata (such as domains) of all data and

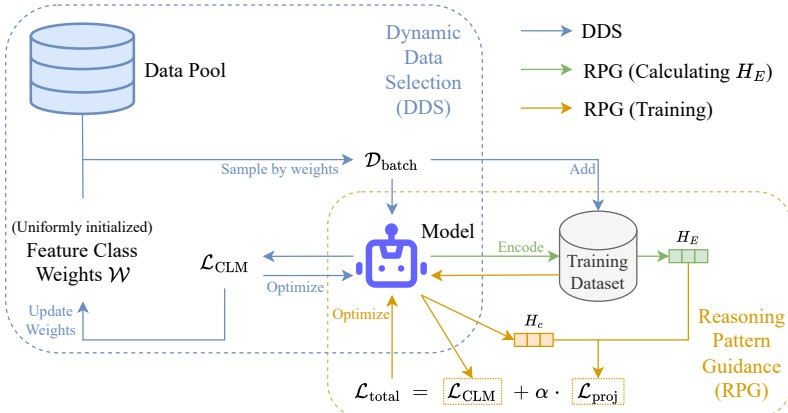

Figure 1: An overview of the DynaGuide framework. For the first training epoch, the DDS module iteratively updates data feature class weights and selects valuable training data (blue paths). During subsequent training, the RPG module calculates the execution step feature vector $H_E$ (green paths) and introduces a projection loss $\mathcal{L}_{\text{proj}}$ to encourage effective and efficient reasoning (yellow paths).

the reasoning trace given by LLM (specifically DeepSeek-R1 (Guo et al., 2025) in our experiments), so we can leverage more comprehensive information for fine-grained data selection. RPG addresses the overthinking problem by incorporating additional guidance during distillation, derived from our systematic analysis of reasoning patterns in knowledge QA tasks. Together, these two components enable efficient and adaptive distillation while improving the distilled model's reasoning capability.

In summary, our work makes the following contributions: (1) We propose Dynamic Data Selection during fine-tuning to better and more efficiently transfer the advanced reasoning ability of LLMs to small models through distillation. (2) We explore the reasoning patterns in knowledge QA and incorporate Reasoning Pattern Guidance into the fine-tuning process to mitigate overthinking and encourage the distilled model to think efficiently and correctly. (3) Comprehensive experiments demonstrate the effectiveness of our framework. Notably, our fine-tuned 7B model can achieve or even exceed the performance of its 32B counterpart. We further provide a systematic analysis of its generalization capability and extensive ablation studies.

## 2 RELATED WORK

**Distillation of Large Language Models** Knowledge Distillation has emerged as a promising approach to transfer the advanced capabilities of LLMs to compact open-source models (Xu et al., 2024b). Early exploration focused on learning specific knowledge from LLMs (Ding et al., 2023), while recent studies attempt to transfer the advanced reasoning capability to small models (Hsieh et al., 2023; Sun et al., 2025), particularly in mathematical and programming domains (Xu et al., 2025; Team, 2025; Labs, 2025). However, the distillation of reasoning-based knowledge QA tasks catches much less attention. Current approaches also exhibit limitations in data curation, including dependence on large-scale datasets (DeepSeek-AI, 2025), reliance on coarse-grained data selection (Muennighoff et al., 2025), and necessity for costly human expert annotations (Yu et al., 2025). Therefore, our work investigates data-efficient distillation through fine-grained data selection in knowledge QA tasks.

**Data Selection** Existing data selection methods are typically designed for training base models with large-scale datasets, rather than targeting specific downstream tasks. Some approaches rely on domain-level loss estimation, even requiring multi-stage processes to train additional proxy models, reference models checkpoints, and optimizers for estimation (Xie et al., 2023; Xia et al., 2023; Jiang et al., 2024b; Luo et al., 2024b), which leads to significant computational cost. Some other selection methods rely solely on intrinsic data properties, without considering the adaptability between model and data or distinguishing between different domains (Qin et al., 2023; Yang et al., 2025b).

**Knowledge QA** As LLM continues to evolve, performance on QA tasks gradually improves, but problems such as hallucinations still exist (Huang et al., 2023; Jiang et al., 2024a; Luo et al., 2024a).

Retrieval-augmented generation (RAG) can be helpful by introducing external knowledge into the context or training objectives (Gao et al., 2023; Asai et al., 2023; Tu et al., 2025). Knowledge-based QA is well suited for testing the model's reasoning ability, as it is a challenging task to reconcile multiple knowledge and reason between input texts (Yang et al., 2018; Geva et al., 2021). Previous works have similarly proposed training to improve reasoning ability on knowledge QA tasks, but require large amounts of labeled or generated data (Xu et al., 2024a; Lyu et al., 2024). Our work focuses on data selection to achieve optimal results with a small amount of data and to maintain the model's ability to generalize, improving model's reasoning ability in both in-domain and out-of-domain knowledge QA tasks.

# 3 DYNAMIC DATA SELECTION DURING FINE-TUNING

---

**Algorithm 1** Dynamic Data Selection during Fine-Tuning

---

**Require:** Training Data Pool $\mathcal{D}$, Model $\theta_0$, Amount of training data $n$, Amount of Warm-up Data $n_w$, Batch Size $n_b$, Upper Threshold $t_u$, Lower Threshold $t_l$.
**Ensure:** Fine-tuned model $\theta$.

1: **function** UPDATEWEIGHTS($\mathcal{W}, \mathcal{D}$, losses)    # Warm-up stage
2:   $\bar{\ell} \leftarrow$ AVERAGE(losses.values())
3:   **for** each $i$ **in** $1, \ldots, |\mathcal{D}|$ **do**
4:     $d_i \leftarrow \mathcal{D}[i]$    # data point
5:     $c_i \leftarrow$ FEATURECLASSOF($d_i$)
6:     $\ell_i \leftarrow$ losses[$i$]
7:     $r_i \leftarrow \ell_i/\bar{\ell}$
8:     **if** $r_i > t_u$ or $r_i < t_l$ **then**
9:       $f_i \leftarrow \max\left(\frac{1}{2}, \frac{2r_i}{r_i+1}\right)$
10:      $\mathcal{W}[c_i] \leftarrow f_i \cdot \mathcal{W}[c_i]$
11:    **end if**
12:   **end for**
13:   $\mathcal{W} \leftarrow$ NORMALIZE($\mathcal{W}$)
14: **end function**

15: **Initialize** Uniform distribution $\mathcal{W}_0$ across all feature classes of data

    # Warm-up stage
16: $\mathcal{D}_{\text{train}} \leftarrow$ SAMPLE($\mathcal{D}, \mathcal{W}_0, n_w$)
17: $\mathcal{D} \leftarrow \mathcal{D} \setminus \mathcal{D}_{\text{train}}$
18: $(\theta_1,$ losses$) \leftarrow$ TRAIN($\theta_0, \mathcal{D}_{\text{train}}$)
19: $\mathcal{W}_1 \leftarrow$ UPDATEWEIGHTS($\mathcal{W}_0, \mathcal{D}_{\text{train}}$, losses)

    # Dynamic Data Selection
20: $i \leftarrow 1$
21: **while** $|\mathcal{D}_{\text{train}}| < n$ **do**
22:   $\mathcal{D}_{\text{batch}} \leftarrow$ SAMPLE($\mathcal{D}, \mathcal{W}_i, n_b$)
23:   $\mathcal{D} \leftarrow \mathcal{D} \setminus \mathcal{D}_{\text{batch}}$
24:   $(\theta_{i+1},$ losses$) \leftarrow$ TRAIN($\theta_i, \mathcal{D}_{\text{batch}}$)
25:   $\mathcal{W}_{i+1} \leftarrow$ UPDATEWEIGHTS(
            $\mathcal{W}_i, \mathcal{D}_{\text{batch}}$, losses)
26:   $i \leftarrow i + 1$
27:   $\mathcal{D}_{\text{train}} \leftarrow \mathcal{D}_{\text{train}} \cup \mathcal{D}_{\text{batch}}$
28: **end while**
    **return** $\theta_i$

---

In this section, we present our dynamic data selection framework for fine-tuning. Our fundamental premise is that distinct data characteristics result in divergent learning dynamics during the fine-tuning process. Certain domains or complexity levels require much exposure for adaptation, while others stimulate the model's capabilities through few appearances. In this paper, we characterize the data from two orthogonal dimensions: (i) domain specificity and (ii) task complexity.

Our dynamic data selection methodology, formalized in Algorithm 1, operates on the principle of continuous weight adaptation during fine-tuning. The framework starts with a warm-up phase and maintains a dynamic weight distribution across data feature classes, which it uses to probabilistically sample each subsequent training batch. This adaptive approach enables the model to automatically prioritize data features that require more attention while maintaining exposure to all the classes.

## 3.1 WARM-UP

The cold start problem poses a significant challenge that purely dynamic data selection may lead to insufficient model understanding of the overall data distribution. Without proper initialization, the weights assigned to initially selected data types could progressively increase, creating a self-reinforcing cycle where these data types continue to be preferentially selected. This phenomenon may result in the neglect of other data feature classes, ultimately reducing training data diversity and compromising the model's generalization capability.

To address this issue, we introduce a warm-up phase prior to dynamic data selection. During this phase, we construct a balanced warm-up dataset by uniformly sampling equal amounts of data from all the classes. This warm-up dataset constitutes 4% of the total selected data, serving to establish a more representative initial data distribution before transitioning to dynamic selection.

## 3.2 Dynamic Selection

Since performing inference on the entire training data pool to identify samples with the highest model uncertainty is computationally prohibitive, our approach dynamically adjusts the weights of data feature classes that result in higher or lower loss in the currently observed batch. This strategy aims to prioritize the selection of such informative samples in subsequent training iterations.

Specifically, our approach calculates the ratio of each sample's loss to the batch's average loss during training. Subsequently, conditioned on this ratio, we implement weight adjustments: for samples with a ratio below a lower threshold $t_l$, we downweight the type to which the sample belongs; conversely, for samples with a ratio above an upper threshold $t_u$, we upweight the corresponding type. These thresholds act as a margin to explicitly separate samples the model finds easy (low loss) from those it finds difficult (high loss), thereby stabilizing the weighting mechanism.

We compute a weight adjustment factor $f$ based on the loss ratio. The underlying principle is to assign larger weight increments to types with higher loss ratio values and larger reductions to those with lower loss ratio values. To mitigate weight explosion or weight disappearance, we require the weight growth rate to be sublinear with respect to the loss ratio. Thus, we adopt a simple rational function with a lower limit for smoother scaling:

$$f = \max\left(\frac{1}{2}, \frac{2r}{r+1}\right), \quad \text{where } r = \frac{\ell_i}{\bar{\ell}_{\text{batch}}}. \tag{1}$$

Here, $\ell_i$ denotes the per-sample loss and $\bar{\ell}_{\text{batch}}$ represents the batch-averaged loss. Such $f$ ensures monotonic yet controlled adjustments, approaching 2 for large $r$ and 0.5 for small $r$, reducing the sensitivity to extreme values.

Upon selecting a predefined number of instances, we terminate the dynamic data selection process. To demonstrate the data efficiency of our method and facilitate a fair comparison with prior work (Muennighoff et al., 2025), we limit the total selected data to 1,000 samples. The fine-tuning procedure consists of 5 epochs, with the dynamic data selection performed exclusively during the first epoch. After that, we train the model on the selected subset for another 4 epochs. Such a procedure ensures consistent evaluation conditions and maintains computational efficiency.

## 4 Incorporate Control of thinking

To systematically analyze the reasoning patterns in knowledge-based question answering tasks, we follow the definition of Chen et al. (2025) to segment reasoning traces into discrete steps using double newline delimiters ('\n\n') and categorize these steps into three distinct types: execution, reflection, and transition. Execution steps perform factual retrieval or concrete computation, reflection steps verify the previous steps, and transition steps bridge two different reasoning paths.

### 4.1 Reasoning Patterns Analysis

Table 1: Analysis of DeepSeek-R1's reasoning patterns in knowledge question answering tasks.

| Metric | Answer Type | |
| --- | --- | --- |
| | Correct | Wrong |
| Average # Tokens | 1804.54 | 1823.11 |
| Execution Steps | 73.10% | 58.40% |
| Reflection Steps | 17.13% | 23.30% |
| Transition Steps | 9.77% | 18.30% |

First, we analyze the model's reasoning patterns in knowledge QA tasks, with particular attention to the correlation between step-type frequencies and task performance metrics. Table 1 presents an analysis of DeepSeek-R1's chains of thought on knowledge QA tasks (on strategyQA (Geva et al., 2021), hotpotQA (Yang et al., 2018) and superGPQA (Du et al., 2025) datasets). We systematically examined the model's performance by quantifying the average token length of reasoning chains, and the distribution of different reasoning step types across both correct and incorrect responses.

Different from the findings of previous work (Chen et al., 2025) in the field of mathematical tasks, we find that in the field of knowledge QA tasks, there is no significant difference in the number of model's thinking tokens when answering correctly and incorrectly. However, our analysis reveals distinct patterns in reasoning step type distributions between correct and incorrect responses. For erroneous answers, we observe a statistically significant decrease in execution-type steps, accompanied by a marked increase in other step types, particularly transition steps. This inverse relationship suggests that excessive reflection and transition steps may disrupt the model's reasoning process, potentially leading to performance degradation. Specifically, the disproportionate growth in meta-cognitive steps appears to compromise the model's ability to maintain focused reasoning.

## 4.2 REASONING PATTERN GUIDANCE

We further perform encoding on the training set to extract the hidden states of the tokens containing '\n\n' as feature vectors of subsequent thinking steps. Our analysis reveals that vectors from deeper layers of model exhibit weak separability when projected into 2D space (more details are provided in Appendix E). This observation suggests stronger separability in the original high-dimensional hidden space. This discovery is also consistent with the conclusions of Chen et al. (2025) on mathematical tasks. Therefore, we propose to apply Reasoning Pattern Guidance (RPG) by introducing a projection loss at the deep layers of the model.

To guide the model's reasoning process during fine-tuning and encourage more execution steps, we propose adjusting the hidden states of tokens containing '\n\n' toward the direction of the average execution-step feature vector, denoted as $H_E$. However, since $H_E$ may evolve during training, we employ an iterative tuning approach to align the representations with the target reasoning trajectory.

Spefically, before each epoch $i$, we perform encoding on the selected training dataset and calculate the average execution-step feature vector $H_E^i$. During training , we introduce a projection loss for all tokens containing '\n\n'. Let $H_c^i$ represent the hidden state of such a token at epoch $i$, and let Sim represents a similarity function such as cosine similarity. The loss encourages alignment between $H_c^i$ and the average execution-step feature vector $H_E^i$ from the previous epoch:

$$\mathcal{L}_{\text{proj}} = \frac{(1 - \text{Sim}(H_c^i, H_E^i))}{2}. \tag{2}$$

If the subsequent step is a reflection step or a transition step, we apply the projection loss, the loss 'pushes' $H_c^i$ toward $H_E^i$, thereby encouraging the model to generate more execution steps. If the subsequent step is already an execution step, the employment of the same projection loss minimizes the semantic drift in the learned representations. This ensures consistent optimization across all reasoning step types, resulting in a unified projection loss.

We jointly optimize both the causal language modeling loss and the projection loss during training. The total loss function is defined as:

$$\mathcal{L}_{\text{total}} = \mathcal{L}_{\text{CLM}} + \alpha \cdot \mathcal{L}_{\text{proj}}, \tag{3}$$

where $\mathcal{L}_{\text{CLM}}$ denotes the standard causal language modeling loss, $\mathcal{L}_{\text{proj}}$ is our proposed projection loss that enforces latent space alignment constraints, and $\alpha$ serves as a balancing hyperparameter. This configuration maintains equilibrium between the two objectives and prevents significant deviation from the original pre-trained model's hidden space.

## 5 EXPERIMENTS

### 5.1 SETUP

**Datasets** Our training dataset consists of the training set of StrategyQA (Geva et al., 2021), the training set of HotpotQA (Yang et al., 2018), and SuperGPQA (Du et al., 2025), aggregating to

118,579 samples that span diverse domains and complexity levels. Then we apply a quality filter retaining only questions that neither Qwen2.5-7B-Instruct nor Qwen2.5-32B-Instruct (Team, 2024) can answer correctly, which is similar to s1 (Muennighoff et al., 2025). This filtering process results in a refined training data pool of 71,662 examples. Then we request the DeepSeek-R1's inference API to generate reasoning traces and answers, which serve as pseudo-annotations for each question. This dataset forms the training data pool for our dynamic data selection framework.

Our methodology focuses specifically on transferring knowledge reasoning capabilities from Reasoning LLMs to small models, rather than context retrieval performance. To maintain this focus, we omit external context in our training examples, requiring the model to rely exclusively on its internal knowledge for reasoning. This design ensures the fine-tuning process specifically enhances the model's inherent reasoning abilities without confounding factors from retrieval augmentation.

**Implementation Details**   We conduct full-parameter supervised fine-tuning (SFT) of models from different series, including Qwen2.5-7B-Instruct (Team, 2024), LLaMA-3.1-8B-Instruct (Grattafiori et al., 2024), Qwen3-4B and Qwen3-8B (Yang et al., 2025a), using a two-phase training approach: (i) During the initial epoch, we perform dynamic data selection described in Section 3 until accumulating a curated set of 1,000 training examples; (ii) For the subsequent four epochs, we train exclusively on the selected subset while incorporating our proposed reasoning pattern guidance (RPG) framework, described in Section 4.2. In our experiment, we set the thresholds $t_l = 0.9$ and $t_u = 1.1$, and the loss weighting hyperparameter $\alpha = 1.0$. For the 28-layer Qwen2.5-7B-Instruct model, we introduce the RPG projection loss at the 20th layer; for both the 32-layer LLaMA-3.1-8B-Instruct and the 36-layer Qwen3-4B and Qwen3-8B models, the RPG projection loss is introduced at the 24th layer. More experimental details can be found in Appendix C.

**Evaluation**   Our evaluation protocol includes both in-domain and out-of-domain datasets. For in-domain evaluation, we assess model performance on the StrategyQA test set and the HotpotQA development set (since the answers of the HotpotQA test set are unavailable). For out-of-domain evaluation, we test the model on some challenging benchmarks: FRAMES (Krishna et al., 2024), GPQA extended set (Rein et al., 2024), and SimpleQA (Wei et al., 2024). Our evaluation covers both context-free question answering and retrieval-augmented generation (RAG) question answering settings. All results report the accuracy rate. More datasets details can be found in Appendix C.1.

**Baselines**   We compare our framework with: (1) **R1-Distill-Qwen-7B** and **R1-Distill-LLaMA-8B** (DeepSeek-AI, 2025): models distilled on 800K data from DeepSeek-R1 based on Qwen and Llama, released by DeepSeek-AI; (2) The foundation model prior to our fine-tuning of each model series; (3) **s1** and **s1 + BF**: a data selection and distillation framework by Muennighoff et al. (2025) but on knowledge reasoning tasks, where BF denotes their proposed test-time scaling technique Budget Forcing; and (4) **Qwen2.5-32B-Instruct** (Team, 2024) (for Qwen2.5 Series only): a model with approximately 4x parameters for cross-scale comparison.

5.2   MAIN RESULTS

The main experimental results are presented in Table 2, highlighting the superior performance of our framework across a wide range of benchmarks. Our proposed distillation framework consistently boosts model performance in all series, delivering significant improvements on nearly every benchmark. Compared to R1-Distill models of similar size, which are fine-tuned on 800K data, our model achieves higher accuracy on most benchmarks. Furthermore, our framework demonstrates better performance than the s1 framework with the same data efficiency. Notably, our fine-tuned Qwen2.5-7B-Instruct model matches or even surpasses the performance of Qwen2.5-32B-Instruct, despite having only about a quarter of the parameters.

Additionally, although our models are trained exclusively on context-free reasoning QA tasks, they exhibit strong generalization to retrieval-augmented generation (RAG) Question Answering tasks that require contextual reasoning. For further details, please refer to Appendix B.

Table 2: Model accuracy on various benchmarks. [c]Question answering tasks with retrieved context (i.e., RAG evaluation) requiring contextual reasoning. **Bold** formatting highlights the top-performing model within the corresponding series.

| Model | Fine-Tuning Data | In-domain Datasets | | | Out-of-domain Datasets | | | |
|---|---|---|---|---|---|---|---|---|
| | | StrategyQA test set | HotpotQA dev | HotpotQA dev[c] | FRAMES[c] | GPQA extended | SimpleQA | SimpleQA[c] |
| R1-Distill-Qwen-7B | 800K | 70.16 | 19.27 | 54.80 | 22.33 | 46.34 | 2.17 | 46.44 |
| R1-Distill-LLaMA-8B | 800K | 71.03 | 20.20 | 55.24 | 23.18 | 50.73 | 2.36 | 48.94 |
| Qwen2.5-7B-Instruct Series | | | | | | | | |
| Qwen2.5-7B-Instruct | - | 68.41 | 15.87 | 46.91 | 20.63 | 31.87 | 2.47 | 47.55 |
| s1-Qwen2.5-7B | 1,000 | 74.09 | 20.77 | 57.92 | 23.54 | 39.74 | 3.05 | 46.51 |
| s1-Qwen2.5-7B + BF | 1,000 | 74.53 | 21.36 | 59.43 | 25.24 | 44.51 | 3.33 | 49.40 |
| **DynaGuide-Qwen2.5-7B** | 1,000 | **77.73** | **24.98** | **61.59** | **28.52** | **47.07** | **3.98** | **54.60** |
| Qwen2.5-32B-Instruct | - | 75.98 | 27.94 | 65.47 | 27.55 | 42.31 | 4.90 | 53.42 |
| LLaMA-3.1-8B-Instruct Series | | | | | | | | |
| LLaMA-3.1-8B-Instruct | - | 70.45 | 24.08 | 54.53 | 20.39 | 35.16 | 3.88 | 47.39 |
| s1-LLaMA-3.1-8B | 1,000 | 74.24 | 25.62 | 58.60 | 25.61 | 39.28 | 3.95 | 48.13 |
| s1-LLaMA-3.1-8B + BF | 1,000 | 74.38 | 27.02 | 60.85 | 26.94 | 46.15 | **4.16** | 51.90 |
| **DynaGuide-LLaMA-3.1-8B** | 1,000 | **77.44** | **28.16** | **63.24** | **27.67** | **48.35** | 4.07 | **55.34** |
| Qwen3-4B Series | | | | | | | | |
| Qwen3-4B | - | 72.49 | 19.31 | 55.22 | 23.79 | 45.06 | 2.50 | 46.63 |
| s1-Qwen3-4B | 1,000 | 72.78 | 20.96 | 64.27 | 26.21 | 49.27 | 2.54 | 50.90 |
| s1-Qwen3-4B + BF | 1,000 | 73.36 | 23.67 | 65.93 | 27.55 | 50.73 | 2.61 | 51.87 |
| **DynaGuide-Qwen3-4B** | 1,000 | **75.69** | **24.19** | **66.27** | **27.67** | **52.01** | **2.75** | **52.71** |
| Qwen3-8B Series | | | | | | | | |
| Qwen3-8B | - | 74.09 | 22.61 | 58.23 | 27.55 | 50.37 | 3.00 | 50.92 |
| s1-Qwen3-8B | 1,000 | 76.57 | 23.09 | 64.75 | 28.76 | 52.20 | 3.07 | 53.33 |
| s1-Qwen3-8B + BF | 1,000 | 78.31 | 23.94 | 66.05 | 30.34 | 53.85 | 3.51 | 54.42 |
| **DynaGuide-Qwen3-8B** | 1,000 | **79.62** | **25.10** | **66.58** | **32.16** | **54.58** | **4.11** | **56.06** |

# 6 ANALYSIS

To thoroughly validate the effectiveness of each component within our framework, we further conduct an extensive set of analytical experiments based on the Qwen2.5-7B-Instruct model series.

## 6.1 ABLATION STUDY OF DYNAMIC DATA SELECTION

Table 3: Performance comparison of different data selection methods. [c]RAG Question answering tasks requiring contextual reasoning. **Bold** formatting highlights the top-performing model. All results report task accuracy. We also report the radius of the 95% confidence interval in the ablation study of the warm-up phase. All models are trained without Reasoning Pattern Guidance.

| Method | In-domain Datasets | | | Out-of-domain Datasets | | | |
|---|---|---|---|---|---|---|---|
| | StrategyQA test set | HotpotQA dev | HotpotQA dev[c] | FRAMES[c] | GPQA extended | SimpleQA | SimpleQA[c] |
| Random | 73.95 | 21.07 | 60.03 | 21.84 | 37.73 | 2.80 | 48.20 |
| Longest | 69.29 | 18.16 | 56.88 | 24.52 | **46.15** | 2.89 | 48.66 |
| s1 | 74.09 | 20.77 | 57.92 | 23.54 | 39.74 | 3.05 | 46.51 |
| w/o Warm-up | 76.61 ± 1.40 | 22.75 ± 1.45 | 59.88 ± 1.89 | 23.14 ± 2.08 | 41.15 ± 2.49 | 2.96 ± 0.14 | 48.55 ± 0.47 |
| DDS (ours) | **77.29** ± 1.03 | **23.92** ± 0.70 | **60.18** ± 1.24 | **24.60** ± 0.97 | 43.28 ± 1.56 | **3.28** ± 0.17 | **49.47** ± 0.57 |

We compared our proposed Dynamic Data Selection framework with several baseline data selection methods: (1) **Random**: Uniform random sampling of 1,000 instances; (2) **Longest**: Selecting samples with the longest R1 reasoning chains; (3) **s1**: the data selection method described by Muennighoff et al. (2025), which comprise two phases - initial uniform sampling of 300 instances, followed by repeatedly difficulty-weighted data selection in a randomly chosen domain, until reaching 1,000 instances. We also conduct an ablation study to show the necessity of the warm-up phase in DDS.

As shown in Table 3, our proposed data selection method achieves significant performance gains across multiple tasks, while baseline methods suffer from uneven data distribution, especially the Longest baseline. Since most training data with the longest reasoning chain originates from reasoning traces on the SuperGPQA dataset, the most difficult training set, this baseline yields huge improvements on the similar GPQA Benchmark, but shows minimal gains on other datasets. Furthermore, incorporating the warm-up phase enhances overall performance and stabilizes training dynamics.

## 6.2 ABLATION STUDY OF REASONING PATTERN GUIDANCE

We evaluate our proposed Reasoning Pattern Guidance framework against several baseline approaches that may facilitate the model's execution step: (1) **Vanilla SFT**: The standard approach where only supervised fine-tuning (SFT) is performed on the selected data; (2) **Exe-only**: A simplified variant where we remove all reflection and transition steps from the training data, retaining only the execution steps; (3) **SEAL** (Chen et al., 2025): An approach that performs targeted modifications to the model's hidden states during decoding to encourage more execution steps.

Table 4: Performance and average thinking token number comparison of reasoning control methods. The average number of thinking tokens is shown in parentheses. **c**RAG Question answering tasks requiring contextual reasoning. **Bold** formatting highlights the top-performing model. All results report task accuracy. All models are trained with Dynamic Data Selection.

| Method | In-domain Datasets | | | Out-of-domain Datasets | | | |
|---|---|---|---|---|---|---|---|
| | StrategyQA test set | HotpotQA dev | HotpotQA dev**c** | FRAMES**c** | GPQA extended | SimpleQA | SimpleQA**c** |
| Vanilla SFT | 77.29 (861.89) | 23.92 (2639.62) | 60.18 (1236.33) | 24.60 (2036.92) | 43.28 (23381.84) | 3.28 (1644.97) | 49.47 (1068.48) |
| Exe-only | 75.25 (793.64) | 22.32 (2341.80) | 60.04 (1260.39) | 24.03 (1779.49) | 43.22 (20991.40) | 2.96 (1722.02) | 50.18 (763.99) |
| SEAL | 76.86 (605.44) | 24.51 (1067.05) | 60.28 (730.00) | 26.82 (1581.05) | 45.79 (10652.20) | 3.65 (1033.96) | 51.69 (682.63) |
| RPG (ours) | **77.73** (788.71) | **24.98** (1991.35) | **61.59** (990.71) | **28.52** (1511.84) | **47.07** (16723.81) | **3.98** (1421.03) | **54.60** (786.53) |

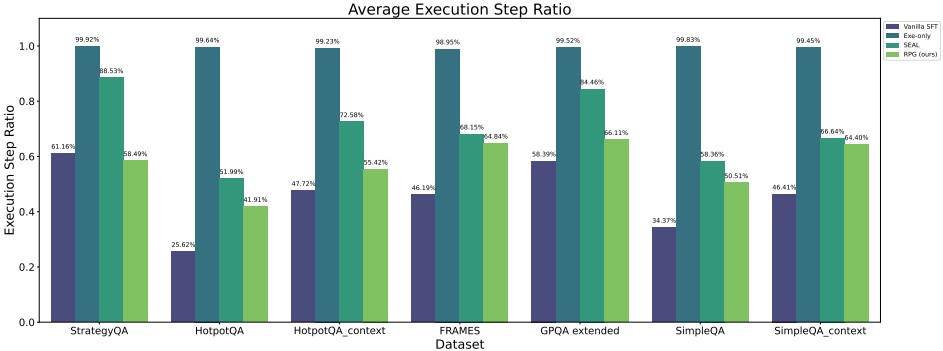

Figure 2: Comparison of the proportion of execution steps of different methods on various datasets. "HotpotQA" and "SimpleQA" denote inputs without external context, while "HotpotQA_context" and "SimpleQA_context" represent RAG tasks with external context input.

As demonstrated in Table 4, our method achieves the most significant performance gains over Vanilla SFT across all evaluated datasets. Notably, the Exe-only variant exhibits degraded performance, which we attribute to the removal of reflection and transition steps. This modification disrupts the coherence of reasoning chains, compromising both semantic integrity and contextual relevance.

We further analyzed the execution step ratios across different methods (Figure 2), revealing two key insights: (1) Exe-only achieves near-complete execution dominance. However, this comes at the cost of semantic coherence, as the removal of reflection and transition steps leads to fragmented reasoning chains and compromised contextual relevance, ultimately impairing task performance. (2) While SEAL demonstrates higher execution rates than our RPG framework, this comes through forced conversion of reflection/transition steps into execution during decoding. In contrast, RPG maintains the model's capacity for necessary reflection and transition while promoting execution during training, achieving superior overall performance through more balanced reasoning processes.

## 6.3 Impact of the Correctness of R1-Responses

Furthermore, we investigate a critical scientific question: *Does the small model primarily acquire factual knowledge through distillation, or does it mainly develop reasoning capabilities?* To examine this distinction, we conduct experiments using two distinct data pools respectively: (i) **R1-correct**: Samples where DeepSeek-R1 provides correct answers, containing accurate reasoning traces; (ii) **R1-wrong**: Samples where DeepSeek-R1 provides incorrect answers, representing cases where the reasoning traces contain erroneous knowledge. This design enables us to distinguish the model's ability to learn reasoning patterns from its capacity to acquire factual knowledge through distillation.

Table 5: Performance comparison of models trained on different training data pools. [c]Question answering tasks with retrieved context (i.e., RAG evaluation) requiring contextual reasoning. All models are trained with Dynamic Data Selection and without Reasoning Pattern Guidance.

| Training Data Pool | Data Size | In-domain Datasets | | | Out-of-domain Datasets | | | |
|---|---|---|---|---|---|---|---|---|
| | | StrategyQA test set | HotpotQA dev | HotpotQA dev[c] | FRAMES[c] | GPQA extended | SimpleQA | SimpleQA[c] |
| R1-correct | 25,725 | 78.31 | 23.59 | 62.04 | 25.36 | 45.24 | 3.77 | 51.62 |
| R1-wrong | 45,937 | 75.40 | 23.36 | 59.22 | 23.30 | 38.28 | 2.61 | 48.75 |
| All | 71,662 | 77.29 | 23.92 | 60.18 | 24.60 | 43.28 | 3.28 | 49.47 |

Our experimental results (Table 5) reveal some key observations: First, models fine-tuned exclusively on incorrect reasoning chains (R1-wrong) still achieve competitive performance. Second, the performance gain from using only correct chains (R1-correct) is marginal compared to training on the complete dataset. These results strongly suggest that the model primarily acquires reasoning capabilities rather than merely memorizing factual knowledge during the distillation process.

## 7 Conclusion and Future Work

In this work, we present DynaGuide, an innovative framework for efficiently distilling the reasoning capabilities of LLMs into more compact and deployable models. First, our proposed Dynamic Data Selection (DDS) provides better data curation than current distillation approaches. Second, the Reasoning Pattern Guidance (RPG) resolves the overthinking issue in LLM-generated synthetic data by optimizing the reasoning process during fine-tuning. Together, these components enable more data-efficient distillation while maintaining the reasoning quality of distilled models. Furthermore, our extensive analysis of data selection and the model's reasoning pattern provides valuable insights for future research, advancing the field of knowledge distillation of LLMs.

Since powerful reasoning LLMs are still relatively rare, and the detailed chains of thought of some advanced models (e.g., OpenAI-o1 and OpenAI-o3) are either not publicly available or only released in a summarized form, we leave further exploration of our method's generalization across diverse teacher models as future work.

## REPRODUCIBILITY STATEMENT

To promote the reproducibility of our work, the datasets, hyperparameters, and experimental details used in our study are provided in Section 5.1 and Appendix C. Upon acceptance of this paper, we will also publicly release our source code and model checkpoints to enhance transparency and enable future research.

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

## A LARGE LANGUAGE MODEL (LLM) USAGE STATEMENT

In accordance with the conference guidelines regarding the use of Large Language Models (LLMs), we hereby disclose the following: After completing the initial draft of this paper, we used an LLM for language polishing and improving grammar. The LLM was not involved in generating any novel research ideas, designing experiments, or creating original scientific content. All research findings, analyses, and conclusions in this paper are entirely the work of the authors. The authors bear full responsibility for the entirety of the paper's content, including any sections that were revised with LLM assistance. The LLM is not listed as an author and did not participate in the conception of research ideas or the substantive scientific writing process.

This disclosure is made to maintain transparency and adhere to the conference's policies regarding the use of LLMs.

## B GENERALIZATION TO RAG TASKS

Since our model is fine-tuned without external context and the RAG task represents an important knowledge-based question answering scenario, we additionally evaluate our framework's generalization capability to the RAG task on hotpotQA, FRAMES, and SimpleQA datasets.

We organize the external context input to the model as follows: (i) For the HotpotQA dataset, the context is contained in the dataset file, and we concatenate several pieces of text in their original order; (ii) For the FRAMES and SimpleQA datasets, the data files contain context URLs. We crawl all Wikipedia URLs and delete irrelevant content such as navigation bars, sidebars, and hyperlinks, retaining the title and body, and then concatenate the texts to form the context according to the order of the URLs in the data file. Since the context length of the model is limited and some context windows need to be reserved for reasoning, we truncate all contexts and only keep the first 8K tokens as context input to the model.

In addition to superior ability to rely on internal knowledge for reasoning, the results in Table 2 on HotpotQA dev$^c$, FRAMES$^c$ and SimpleQA$^c$ show that the model fine-tuned by our framework also has excellent contextual reasoning capabilities.

Table 6: Comparison of average reasoning chain length (number of tokens) between s1-Qwen2.5-7B and our DynaGuide-Qwen2.5-7B model under different input settings. "HotpotQA" and "SimpleQA" datasets denote inputs without external context, while "HotpotQA$^c$" and "SimpleQA$^c$" datasets represent RAG tasks with context.

| Model | Datasets | | | |
| --- | --- | --- | --- | --- |
| | HotpotQA | HotpotQA$^c$ | SimpleQA | SimpleQA$^c$ |
| s1-Qwen2.5-7B | 4051.83 | 2107.65 | 2874.94 | 1763.40 |
| DynaGuide-Qwen2.5-7B | 2844.78 | 1256.22 | 2030.04 | 1123.62 |

Furthermore, we evaluate the models' reasoning efficiency on RAG tasks. As illustrated in Table 6, the introduction of external contextual information significantly reduces required reasoning steps by providing supplementary evidence. In addition to the smaller absolute number of reasoning tokens, our DynaGuide-Qwen2.5-7B model reduces the average number of reasoning tokens by 50.25% compared to scenarios without external context, while s1-Qwen2.5-7B achieves a relative reduction of only 43.32%, indicating that our model can perform contextual reasoning more efficiently.

## C EXPERIMENTAL DETAILS

### C.1 DATASETS

We provide a brief description of the datasets used in this work. All these datasets are in English.

Our training dataset consists of:

- The training set of StrategyQA (Geva et al., 2021), a question answering (QA) benchmark that requires multiple reasoning steps for each question. The questions are short but span diverse topics.
- The training set of HotpotQA (Yang et al., 2018), a QA dataset that requires reasoning over multiple supporting documents, which include factual knowledge.
- The SuperGPQA dataset (Du et al., 2025), a QA benchmark covering 285 subjects to test the model's graduate-level knowledge and reasoning capabilities.

Our in-domain evaluation benchmark includes:

- The test set of StrategyQA (Geva et al., 2021).
- The development set of HotpotQA (Yang et al., 2018), since the answers of its test set are unavailable.

And our out-of-domain evaluation benchmark includes:

- The FRAMES dataset (Krishna et al., 2024), a QA benchmark to test model performance in RAG scenarios. It requires multi-step reasoning over factual information from multiple sources.
- The GPQA extended set (Rein et al., 2024), a challenging graduate-level QA benchmark in biology, physics, and chemistry domains. We use the extended set in our evaluation.
- The SimpleQA (Wei et al., 2024) dataset, a QA benchmark including short questions that require factual retrieval and reasoning ability, covering a wide range of topics.

## C.2 TRAINING DETAILS

We list the details of training hyperparameters in Table 7.

Table 7: Training hyperparameters.

| Hyperparameter | Value |
|---|---|
| Batch size | 8 |
| Number of machines | 1 |
| Number of processes | 8 |
| Training epochs | 5 |
| Training steps | 625 |
| Learning rate | 5e-6 |
| Optimizer | AdamW ($\beta_1 = 0.9$, $\beta_2 = 0.999$) |
| Weight Decay | 0.01 |
| Scheduler | Cosine schedule |
| Warmup Steps | 62 |
| ZeRO optimization stage | 3 |
| Mixed precision | bf16 |
| $t_l$ | 0.9 |
| $t_u$ | 1.1 |
| $\alpha$ | 1.0 |

## C.3 PROMPTS

We use the prompt shown below to request DeepSeek-R1 API for its reasoning trace.

```
You are a helpful assistant. You will be given a question. You need to answer
the question by reasoning step by step. In the end, output the final answer in
a new line with the prefix "Final answer:". The final answer should be yes or
no, a choice letter, or a short phrase, without further explanations.
Question: {Question}
Options: {Options} (if there are choices for the question)
```

We use the prompt shown below to evaluate all the models across all the datasets.

```
You are {Qwen / LLaMA}, created by {Alibaba Cloud / Meta}. You are a helpful
assistant. You will be given a question. You need to answer the question by
reasoning step by step. In the end, output the final answer in a new line with
the prefix "Final answer:". The final answer should be yes or no, a choice
letter, or a short phrase, without further explanations.
Context: {Context} (if there exists a context for the question)
Question: {Question}
Options: {Options} (if there are choices for the question)
```

## D  CASE STUDY

We show the output of original Qwen2.5-7B-Instruct and that of our DynaGuide-Qwen2.5-7B on a question from the GPQA dataset in Table 8. Qwen2.5-7B-Instruct lists all the systems and claims all of them can coexist in a multi-star system directly, thus giving a wrong answer, which is not even included in the choices. DynaGuide-Qwen2.5-7B gives its reasoning step by step (we omit many reasoning steps here) and finally outputs the correct answer. Furthermore, it thinks the possible systems that can coexist are the second, third, and fourth, which accurately matches the ground truth explanation in the dataset. This case study demonstrates the effectiveness of our framework.

## E  T-SNE VISUALIZATION OF FEATURE VECTORS OF DIFFERENT THINKING STEP

We provide the t-SNE visualization (Van der Maaten & Hinton, 2008) of feature vectors of different thinking step types in the Qwen2.5-7B-Instruct model in Figure 3. It can be observed that at the deep layer of the model, especially the 20th layer, the vectors show weak separability after projection to 2D space, which implies they are more separable in the original high-dimensional hidden space. This justifies our employment of the projection loss in Reasoning Pattern Guidance (RPG) at the 20th layer when fine-tuning Qwen2.5-7B-Instruct model. Similar phenomena can also be observed in other model series.

Table 8: Case Study

| **Question**: The majority of stars in our Galaxy form and evolve in multi-stellar systems. Below are five potential multi-star systems that are presented. How many of these systems can coexist? 

 W Virginis type star, G2V, M4V, RGB star(1.5Msun) 
 WD (B5 when in the MS) and A0V 
 G2V, K1V, M5V 
 DA4, L4 
 WD (MS mass of 0.85Msun), K3V, A star with a mass of 0.9Msun in the MS. 
 **Options**: A. 3  B. 4  C. 1  D. 2 |
|---|
| **Answer**: A. 3 |

| Qwen2.5-7B-Instruct | DynaGuide-Qwen2.5-7B |
|---|---|
| To determine how many of these systems can coexist, we need to understand what a multi-star system is and the characteristics of the stars listed. 
 1. **W Virginis type star, G2V, M4V, RGB star(1.5Msun)**: This system includes a variable star, a main-sequence star, and a red giant. These can coexist in a multi-star system. 
 2. **WD (B5 when in the MS) and A0V**: This system includes a white dwarf and a main-sequence star. These can coexist in a multi-star system. 
 3. **G2V, K1V, M5V**: This system includes three main-sequence stars. These can coexist in a multi-star system. 
 4. **DA4, L4**: This system includes a white dwarf and a brown dwarf. These can coexist in a multi-star system. 
 5. **WD (MS mass of 0.85Msun), K3V, A star with a mass of 0.9Msun in the MS**: This system includes a white dwarf, a main-sequence star, and another main-sequence star. These can coexist in a multi-star system. 
 Each of the five systems listed can coexist in a multi-star system. Therefore, the answer is: 
 Final answer: 5 | think 
 Okay, let's try to figure this out. The question is about how many of the five multi-star systems can coexist. The options are A, B, C, D, and the answer choices are numbers. Wait, but the user didn't provide options in the original question. Hmm, maybe the options are part of the question here. Let me check again. 
 Wait, the user wrote: "The majority of stars in our Galaxy form and evolve in multi-stellar systems. Below are five potential multi-star systems that are presented. How many of these systems can coexist?" 
 Then the five systems are listed. And the options are A.3 B.4 C.1 D.2. So the options are those four letters. So the answer is one of those. Let me read the systems again. 
 The first system is W Virginis type star, G2V, M4V, RGB star(1.5Msun). I'm not sure what a W Virginis star is. Maybe a variable star? But I don't recall them forming multi-star systems. G2V and M4V are spectral types, which are single stars. So maybe this system can't coexist. 
 The second system is WD (B5 when in the MS) and A0V. WD might be a white dwarf, but B5 and A0 are main sequence stars. Wait, white dwarfs form when main sequence stars die. So maybe a system where a white dwarf is part of a binary with another main sequence star. That could happen. So maybe this one can coexist. 
 The third system is G2V, K1V, M5V. All main sequence stars. Maybe they form a triple system. 
 ... 
 In summary, the possible systems that can coexist are the second, third, and fourth. So three. Therefore, the answer is A.3. 
 answer 
 Final answer: A |

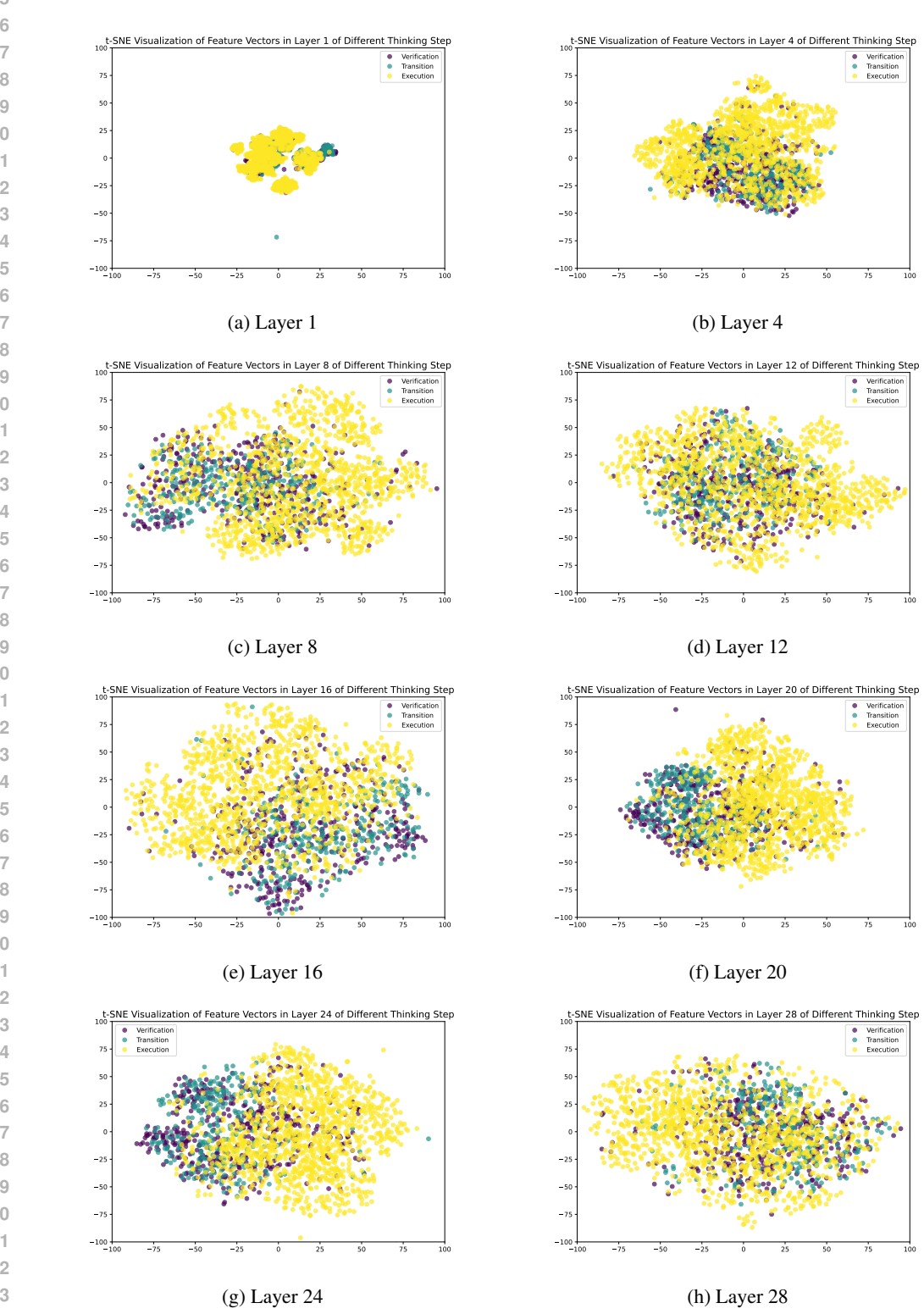

(a) Layer 1

(b) Layer 4

(c) Layer 8

(d) Layer 12

(e) Layer 16

(f) Layer 20

(g) Layer 24

(h) Layer 28

Figure 3: t-SNE Visualization of Feature Vectors of Different Thinking Step in Qwen2.5-7B-Instruct Model across Different Layers

