# OpenReview forum: "Distill Models by Aptitude: Efficient Reasoning Capability Distillation via Adaptive Data Curation and Overthinking Mitigation"
_ICLR.cc/2026/Conference — Submitted to ICLR 2026_

### Official Review · Reviewer_9ndd · 2025-10-28

**Soundness:** 3
**Presentation:** 3
**Contribution:** 2
**Rating:** 6
**Confidence:** 4

**Summary:**

In this paper, the authors proposed two technologies to improve the knowledge-based reasoning distillation tuning. For data selection, a dynamic weight-based sample method is proposed. For reasoning pattern learning, it analyzes the original reasoning pattern and uses a project align loss to guide the learning process. Experiments have demonstrated the effectiveness.

**Strengths:**

The method proposed in the paper has a good effect and provides some inspiration for the inference analysis of the R1 model in the community.

**Weaknesses:**

1. The method requires a more detailed introduction:

1-1. How to sample based on the weights of different categories？

1-2. How are categories divided? Will the fine-grained classification of different categories affect the effectiveness and efficiency of the algorithm? How to deal with the problem of multiple labels in many data classifications？

2. The experimental part needs further improvement.

2-1. It is recommended to provide Table 2 with the performance of the model trained with full data, and compare the performance of the model trained with the sampled data, to more effectively demonstrate the effectiveness of the sampled data.

2-2. It is recommended to compare more sampling methods. Here are some examples that are not necessarily necessary to compare [1] [2].
[1]. Self-Evolved Diverse Data Sampling for Efficient Instruction Tuning
[2]. What Makes Good Data for Alignment? A Comprehensive Study of Automatic Data Selection in Instruction Tuning

2-3. The 32B model in Table 2 is a chat model. It is recommended to compare it with QwQ-32B, which can better improve the ability of the trained 7B model.

**Questions:**

Additional questions:

1. Will the proportion of warm data affect the distribution or quality of sampled data? Do we need corresponding experiments to verify this?

2. Has category balance also been taken into account when sampling by length in Section 6.1?

3. The proposed mapping loss and SFT training loss seem to conflict. Although the features may be closer to the execution step, SFT training still predicts the current token as the data itself. The comparison in Figure 2 also illustrates this point, as the proposed method has limited improvement in the execution step ratio compared to the other two methods. Similarly, does this also indicate that the more execution steps, the better? There may be a reasonable range.

---

> ### Author Response · Authors · 2025-11-24
> **Author Response to Reviewer 9ndd (Part 1/3)**
>
> We sincerely appreciate the reviewer's detailed feedback and constructive suggestions. We are committed to addressing these points to improve the quality of the paper.
>
> ## Response to Weakness 1-1
> The weight vector $W$ is a normalized vector that forms a probability distribution over all the categories. The weight of each category represents its probability of being selected. First, we sample a category $c$ based on this distribution. Then we sample a specific data instance uniformly at random from the pool of all available samples belonging to category $c$ and add it to the training batch.
>
> ## Response to Weakness 1-2
> **1. How are categories divided?**
>
> The categories are divided based on the **Cartesian product** of available metadata attributes, e.g. domain and complexity in our experiment. For example, if we have domains $D=\{\text{Science, History, ...}\}$ and complexity levels $C=\{\text{Easy, Hard, ...}\}$, we define the categories as combinations like $\text{(Science, Easy)}$, $\text{(Science, Hard)}$, $\text{(History, Easy)}$, $\text{(History, Hard)}$, etc.
>
> **2. About Fine-grained Classification**
>
> Effectiveness: **Finer granularity allows for more precise targeting** of the model's weaknesses. However, if the classes are too fine-grained, the number of samples per class decreases. Such **data sparsity** may lead to high variance in the weight estimation of each category.
>
> Efficiency: The cost of maintaining and updating the weight distribution is negligible compared to the cost of training LLMs. Therefore, the algorithm **remains efficient** even with a large number of fine-grained categories.
>
> **3. About Multi-label Data**
>
> This is an interesting question. A natural idea to extend our algorithm to handle multi-label scenarios is to **decompose the weight adjustment factor**. For example, if an instance $d$ belongs to $n$ categories $c_1$, $c_2$, $...$, $c_n$, and the algorithm calculates a total weight adjustment factor $f$ for this instance. Then, we can allocate the adjustment factor by updating the weight of each associated category by $\sqrt[n]{f}$.
>
> ## Response to Weakness 2-1
> We thank the reviewer for this suggestion. We would like to clarify the rationale behind our experimental design.
>
> While the s1 baseline [3] focused on Math reasoning tasks, it has been broadly validated that **high-quality subsets can match or even outperform full-data training** across various domains [4][5]. We adapted the s1 methodology to the QA domain to serve as **a strong, data-efficient baseline**. Although we could not perform full-data training due to the prohibitive computational costs of massive QA datasets within the rebuttal period, the experimental results in Table 2 show that **our method consistently outperforms the s1 baseline** under the exact same data budget. This strongly suggests that our approach is highly effective and is able to match full-data training with high efficiency, which is the core contribution of our work.
>
> **Due to character limitations, our response continues in the following comment.**

---

> ### Author Response · Authors · 2025-11-24
> **Author Response to Reviewer 9ndd (Part 2/3)**
>
> Continuing from the previous comment, here we provide our responses to the rest questions.
>
> ## Response to Weakness 2-2
>
> We thank the reviewer for referring us to these relevant works. We have carefully reviewed the suggested papers and appreciate their contributions to the field of data selection.
>
> As the reviewer kindly noted that these comparisons are not strictly necessary, while we acknowledge the value of these methods, we respectfully omitted them from the direct baseline comparison because they operate under a fundamentally different **computational paradigm**. This decision is primarily based on the **fundamental difference in computational efficiency and resource requirements**, which is the central focus of our research.
>
> **(1)** The method in [1] relies on iterative data selection to maximize the distance relative to the current training set in the embedding space. Therefore, it needs to **compute and update the embeddings for the entire candidate data pool at every iteration** to measure these distances. This introduces massive computational overhead compared to our approach, making it less suitable for the resource-constrained scenarios we target.
>
> **(2)** The data selection pipeline in [2] requires a complex data preparation phase, including **heavy data augmentation** on the seed dataset, and the **training of two separate LLaMA-2-7B models** to predict the quality and complexity of the samples respectively. The computational cost of the data preparation phase even exceeds that of the whole fine-tuning / distillation phase. This **conflicts with our goal of achieving performance gains efficiently**.
>
> We agree that comparison with more baselines will strengthen our paper. However, as discussed in Section 1, **many existing works rely on high computational cost or human experts** annotation and selection. Our current baseline, s1, is a strong, meaningful benchmark that shares the same goal of improving reasoning performance under a fixed budget, and the results in Table 2 are sufficient to demonstrate our contribution within this efficiency-focused context.
>
> ## Response to Weakness 2-3
>
> We acknowledge that QwQ-32B is a specialized reasoning model and would serve as a stronger upper-bound reference. However, we would like to clarify the role of the Qwen2.5-32B-Instruct model in Table 2. In our experiments, this model was **not employed as the teacher model** for distillation or data generation. Instead, it serves as a **strong baseline** from the same model family (Qwen2.5) to demonstrate the effectiveness of our approach. **Our trained 7B model is capable of outperforming the much larger 32B model from the same family**. This highlights the efficacy of our distillation process.
>
> ## Response to Question 1
>
> We thank the reviewer for this insightful question regarding the impact of the warm-up data proportion.
>
> To provide a comprehensive answer, we conducted additional ablation studies varying the warm-up proportion (0%, 4%, 16%, and 40%). Since it is hard to quantify data quality, we report the **average number of reasoning tokens** to reflect the quality. We analyzed the **sampled data distribution** across three source pools (StrategyQA, HotpotQA, SuperGPQA), the **average number of reasoning tokens** in the final sampled training dataset, and the **final performance on HotpotQA and GPQA** datasets.
>
> **Table: Effects of Warm-up Data Proportion.**
> | Warm-up Data Proportion | Sampled Data Distribution (StrategyQA / HotpotQA / SuperGPQA) | Avg. Reasoning Tokens  | HotpotQA | GPQA |
> |:--:|:--:|:--:|:--:|:--:|
> | 0% | 21.2% / 38.6% / 40.2% | 1463.2 | 22.75 | 41.15 |
> | 4% | 8.2% / 39.5 / 52.3% | 2154.5 | 23.92 | 43.28 |
> | 16% | 11.9% / 35.5% / 52.6% | 2007.5 | 23.52 | 44.14 |
> | 40% | 16.1% / 39.9% / 44.0% | 1664.9 | 22.89 | 43.04 |
>
> Our key observations are:
> - **Without warm-up (0%)**, the DDS module lacks prior knowledge, resulting in a relatively **balanced but suboptimal distribution** (high proportion of easier StrategyQA samples). **The CoT is also shorter**.
> - **With a small warm-up data proportion (4% and 16%)**, the DDS module can **upweight** the most challenging SuperGPQA dataset and **downweight** data from StrategyQA. This shift also yields **longer reasoning chains** and **optimal performance**.
> - **Increasing the warm-up data proportion to 40%** causes a **degradation**. The randomly pre-selected warm-up data begins to dominate the training dataset and **dilute the impact of our DDS algorithm**. Consequently, the proportion of StrategyQA rises and the proportion of SuperGPQA drops, resulting in **shorter reasoning chains** and **lower final performance**.
>
> These observations indicate that our DDS algorithm successfully identifies data characteristics that are beneficial for improving reasoning capabilities and adaptively samples data from these classes.
>
> **Due to character limitations, our response continues in the following comment.**

---

> ### Author Response · Authors · 2025-11-24
> **Author Response to Reviewer 9ndd (Part 3/3)**
>
> Continuing from the previous comment, here we provide our responses to the rest questions.
>
> ## Response to Question 2
>
> No, the "Longest" baseline in Section 6.1 does not take category balance into account. It selects data **strictly based on the length of the reasoning chains**.
>
> As discussed in Section 6.1, samples from the most challenging SuperGPQA dataset tend to have longer CoT. Consequently, the "Longest" strategy implicitly biases the selection heavily towards SuperGPQA and achieves high performance on the GPQA benchmark which is similar to the SuperGPQA dataset.
>
> ## Response to Question 3
>
> We thank the reviewer for this insightful observation regarding the relationship between the projection loss and SFT loss, as well as the interpretation of the execution step ratio.
>
> **1. The Conflict between Losses**
>
> The interaction between $\mathcal{L}\_{\text{CLM}}$ and $\mathcal{L}\_{\text{proj}}$ is not a harmful conflict, but acts as a necessary **regularization mechanism**. If we only minimize $\mathcal{L}\_{\text{proj}}$, the model would collapse into generating only execution steps and lose the linguistic coherence. And if we only optimize $\mathcal{L}\_{\text{CLM}}$, the model will perfectly mimic the teacher model's behavior, including the "overthinking" defect. By optimizing both, the model learns to **think coherently** and **not to overthink**.
>
> **2. The More Execution Steps does not Mean the Better**
>
> As discussed in Section 6.2, **a robust reasoning process requires a healthy mix of execution, reflection, and transition**. While we want to reduce excessive reflection and transition (overthinking), some reflection and transition are still necessary for planning and error correction. **A reasonable range** of execution step ratio leads to a **more effective CoT** which prevents overthinking but does not eliminate necessary reasoning steps.
>
> ## References
>
> [1] Wu, Shengguang, et al. "Self-evolved diverse data sampling for efficient instruction tuning." arXiv preprint arXiv:2311.08182 (2023).
>
> [2] Liu, Wei, et al. "What Makes Good Data for Alignment? A Comprehensive Study of Automatic Data Selection in Instruction Tuning." The Twelfth International Conference on Learning Representations.
>
> [3] Muennighoff, Niklas, et al. "s1: Simple test-time scaling." Proceedings of the 2025 Conference on Empirical Methods in Natural Language Processing. 2025.
>
> [4] Ye, Yixin, et al. "Limo: Less is more for reasoning." arXiv preprint arXiv:2502.03387 (2025).
>
> [5] Chen, Lichang, et al. "Alpagasus: Training a better alpaca with fewer data." arXiv preprint arXiv:2307.08701 (2023).

---

### Official Review · Reviewer_ywwV · 2025-10-31

**Soundness:** 2
**Presentation:** 1
**Contribution:** 2
**Rating:** 2
**Confidence:** 4

**Summary:**

The authors propose DynaGuide, a distillation framework which aims for more efficient distillation of knowledge from LLMs to SLMs and proposes a fix to mitigate overthinking. The fix for overthinking is called reasoning pattern guidance (RPG) and is an additional loss term to encourage the hidden state of thinking tokens to align with non-thinking output tokens.

The efficient distillation is performed by a dynamic data selection (DDS) which the authors put forward. The authors assume that each data-point belongs to a particular class or domain. These domains are dynamically weighted according to their loss, similar to previous domain re-weighting schemes such as Doremi [1].

The authors make the interesting observation that incorrect Q&A answers have fewer execution steps than for correct answers. (I have reservations with this analysis, see the weaknesses section). As such the authors propose encouraging more execution steps during finetuning with a Reasoning Pattern Guidance loss term which encourages the hidden state vectors of the for reflection and transition steps to align their directions closer to the average hidden state from all the tokens in an execution step. The authors claim that this encourages more execution steps which are a feature of correctly answered questions.

On a variety of Q&A tasks the authors show that distilling DeepSeek-R1 answers into SLMs with their methods gives overall the best performance. The authors also ablate their data selection method and compare it to various methods for encouraging execution only steps.


[1] Xie, Sang Michael, et al. "Doremi: Optimizing data mixtures speeds up language model pretraining." Advances in Neural Information Processing Systems 36 (2023): 69798-69818.

**Strengths:**

* The observation that the number of tokens does not increase for knowledge question and answers tasks for correct and wrong answers in Table 1 is very interesting. The classification into different types of steps: execution, reflection and transition is also interesting, however requires elaboration (noted in the weaknesses section).
* The DDS method for selecting data for distillation makes a lot of sense and empirically shows strong performance versus baselines like s1 [1]. However is it limited to datasets which contain different classes?

[1] Muennighoff, Niklas, et al. "s1: Simple test-time scaling." arXiv preprint arXiv:2501.19393 (2025).

**Weaknesses:**

My main issue is with the presentation of the paper.
* There are many parts of the paper which are too vague:
    * Lines 144-148. What are divergent learning dynamics? “Certain domains or complexity levels require much exposure for adaptation”, what are domains or complexity levels, what do you mean by exposure?
    * Lines 202-203: how have you categorized the step types? By hand? using a classifier? There is no description of what these three different step types are or examples of these step types.
    * I cannot read the legend and labels in Figure 2.
    * Lines 233-239: there seem to be some very important observations which motivate RPG here but they are not developed well enough or details are in the appendix. For example: the weak separability is with respect to the  ‘\n\n’ tokens and what other tokens? Randomly sampled tokens from other parts of the same sequence?
* Notation is introduced which is not explained. For example what is $W$, $W_0$ and $c_i$ in Algorithm 1?
* Awkward grammar:
    * The title in Section 4.
    * Line 365: “we also report the radius of 95%”, what is a radius in this context?
* Simple spelling mistakes e.g. line 244 - “Spefically”.
* In some rows in Table 3 you use means and standard errors however in others you use a single number. If possible stick to means and standard errors everywhere, although I understand that this requires more GPU resources.



I do not agree with the reasoning in lines 226-230, I could be wrong though. The authors argue that because there is a decrease in the number execution steps and an increase in reflection and transition steps that this **causes** the model’s reasoning process to fail and hence the wrong answer is obtained. Is this not a case of a correlation being misinterpreted as causation? I would have liked to see more diagnosis here. Maybe the LM doesn’t know the answer and so this results in increased reflection and transition steps? This hypothesis is easily verifiable with some made up knowledge questions and answers. I’m not convinced by the reasoning here and this puts into question the entire RPG method.

**Questions:**

* Can you perform dynamic selection with datasets which do not contain different defined domains or classes for example GSM8k?
* What value of $\alpha$ do you use? I would like to see an analysis of how the strength of the RPG loss affects the number of execution steps.

---

> ### Author Response · Authors · 2025-11-24
> **Author Response to Reviewer ywwV (Part 1/2)**
>
> We thank the reviewer for identifying these specific areas where the manuscript lacked clarity or the presentation can be improved. We address all the points below.
>
> ## Response to Weakness 1
> 1. **Learning dynamics** describe how the learning of specific training examples influences the model’s predictions on other examples [1]. "Divergent" means the influence differs significantly based on the characteristics of the data. For example, **simple patterns can be learned quickly**, with the loss dropping rapidly in training, while **complex patterns can make the model struggle**, resulting in a loss that decreases much more slowly.
>    "**Domains**" refers to the specific knowledge categories involved in the questions, such as Science, History, Law, Literature and Arts, etc. "**Complexity Levels**" refers to the difficulty of the question. In our framework, this is determined either by dataset annotations or proxied by the length of the CoT generated by the teacher model.
>    "**Exposure**" refers to the model's contact with data samples during the training phase. And "more exposure" means that the model needs more training samples to learn certain patterns.
> 2. As discussed in Section 4, we employ **a rule-based keyword matching approach to categorize steps**, following the methodology established by [2] (cited in the first paragraph of Section 4). **The description of each step type is as follows**, which is the same as that in [2]:
>    - **Execution thoughts**, where the model analyzes the problem and solves it step by step;
>    - **Reflecting thoughts**, where the model pauses the reasoning process to verify its steps;
>    - **Transition thoughts**, where the model shifts its reasoning flow and rethinks the problem from a different perspective.
> 3. We will improve the readability of Figure 2, which is reduced due to space constraints.
>    Specifically, Figure 2 plots the Average Execution Step Ratio (Y-axis) across different datasets (X-axis). The four bars for each dataset represent four methods: Vanilla SFT, Exe-only, SEAL, and RPG (Ours).
>    We will regenerate this figure with larger fonts to ensure all labels and legends are legible.
> 4. To clarify, the separability is analyzed **among the representations of the step-delimiter tokens ('\n\n') themselves**, grouped by their step types. We do not compare '\n\n' tokens with random tokens. Specifically, we extract the hidden state of the '\n\n' token at the beginning of each step. We observe that these delimiter representations naturally cluster into three distinct groups corresponding to Execution, Reflection, and Transition steps, motivating us to design our RPG projection loss.
>
> ## Response to Weakness 2
> We will explicitly define these notations in Algorithm 1 in the revised version. We now clarify their definitions below:
>
> -   **$W$:** Represents the **dynamic weight vector** maintained over all distinct data feature classes. It determines the sampling probability for each class in the subsequent iteration.
> -   **$W_0$:** Represents the **initial weight vector**. As indicated in Line 15 of Algorithm 1, it is initialized as a uniform distribution to ensure balanced exploration at the start of training.
> -   **$c_i$:** Denotes the specific **data feature class** to which the data point $d_i$ belongs.
> -   **$l_i$:** Denotes the training **loss** computed for the individual data point $d_i$ by the model.
> -   **$r_i$:** Represents the **loss ratio**, calculated as $r_i = \ell_i / \bar{\ell}$, where $\bar{\ell}$ is the average loss of the current batch. This metric quantifies the relative difficulty of sample $d_i$.
> -   **$f_i$:** Represents the **weight adjustment factor** for class $c_i$. It is derived from the loss ratio $r_i$ via the weighting function (Equation 1) and determines how much the weight for class $c_i$ should be increased or decreased based on the current model performance.
>
> ## Response to Weakness 3
> Thank you for pointing out these phrasing issues.
>
> 1. We will rename Section 4 to "Controlling Reasoning Process via Reasoning Pattern Guidance".
> 2. We would like to clarify that by "radius," we meant the **half-width** (or the **margin of error**) of the 95% confidence interval. We will rewrite this sentence to use standard statistical terminology in the revised version.
>
> ## Response to Weakness 4
> Thank you for pointing out the typo. We will correct the spelling of "specifically" in the revised version.
>
> ## Response to Weakness 5
> We would like to clarify that we report confidence intervals specifically for the Warmup ablation rows because the distinct purpose of this ablation study is to **demonstrate the necessity of the warmup phase for training stability**. As shown in Table 3, **removing the "Warmup"** strategy leads to not only **a performance drop** but also **higher variance (instability)** in the results.
>
> **Due to character limitations, our response continues in the following comment.**

---

> ### Author Response · Authors · 2025-11-24
> **Author Response to Reviewer ywwV (Part 2/2)**
>
> Continuing from the previous comment, here we provide our responses to the rest questions.
>
> ## Response to Reviewer's disagreement with the reasoning in lines 226-230
> We appreciate the reviewer's insightful critique of our reasoning.
>
> We agree that in some cases, increased reflection stems from a lack of knowledge. However, the reverse is also observed: **structural redundancy can prevent the effective reasoning and application of knowledge**. In some cases, **the model possesses the necessary knowledge** and even gets the correct final answer early in the reasoning process, **but continues generating excessive and redundant thought sequences** [2][3]. The model can be **trapped in redundant verification loops** and **fails to trigger the necessary execution steps** to finalize the result [4].
>
> **Such overthinking is another prevalent failure mode**. And our RPG method is designed to **mitigate this failure mode**, although **there might be other failure modes**, such as the one mentioned by the reviewer.
>
> As shown in our main results (Table 2) and ablation study (Table 4), applying RPG to encourage execution steps significantly improves the final accuracy. This implies that the model indeed sometimes **struggles due to overthinking**, and **the RPG method indeed mitigates this**.
>
> We will revise Lines 226-230 to present this more nuanced view. Thanks again for your insightful review.
>
> ## Response to Question 1
>
> We thank the reviewer for this question regarding the generalization of our method. The answer is yes. Our Dynamic Data Selection (DDS) framework is **designed to be flexible and does not rely on explicit domain labels**.
>
> For example, in datasets like GSM8k, we can utilize the length of the CoT generated by the teacher model as a quantifiable proxy for problem difficulty or complexity. Then we can define data feature classes based on the proxied complexity.
>
> We would like to clarify that our framework is **agnostic to the specific definition of feature classes**. It effectively supports categorization based on a single dimension (e.g., complexity derived from reasoning length) or multidimensional attributes, making it adaptable to datasets like GSM8k that lack explicit domain labels.
>
> ## Response to Question 2
>
> We thank the reviewer for this constructive suggestion.
>
> As detailed in Appendix C.2 of the submitted paper, we use $\alpha=1.0$ for our experiments.
>
> Following your suggestion, we have conducted **an additional ablation study** to analyze how $\alpha$ influences the ratio of execution steps and final performance. We trained another two models based on Qwen2.5-7B-Instruct backbone with $\alpha=0.5$ and $\alpha=2.0$ respectively, and evaluated them on the HotpotQA (dev split) and GPQA datasets.
>
> **Table: Effect of RPG Projection Loss Weight ($\alpha$) on Execution Step Ratio and Performance.**
> | Dataset | Method | Average Execution Step Ratio (%) | Accuracy (%) |
> |:--:|:--:|:--:|:--:|
> | **HotpotQA_dev** | Vanilla SFT | 25.62 | 23.92 |
> |  | RPG ($\alpha=0.5$) | 40.32 | 24.48 |
> |  | RPG ($\alpha=1.0$) | 41.91 | 24.98 |
> |  | RPG ($\alpha=2.0$) | 44.07 | 25.23 |
> | **GPQA** | Vanilla SFT | 58.39 | 43.28 |
> |   | RPG ($\alpha=0.5$) | 61.58 | 44.69 |
> |   | RPG ($\alpha=1.0$) | 66.11 | 47.07 |
> |   | RPG ($\alpha=2.0$) | 69.63 | 46.15 |
>
> There is a clear **positive correlation** between $\alpha$ and the average execution step ratio. Increasing $\alpha$ consistently encourages the model to generate more execution steps. This confirms that **RPG effectively controls the reasoning patterns** as intended.
>
> **The model's performance also improves** as we introduce the RPG projection loss compared to Vanilla SFT. However, when it comes to $\alpha=2.0$, the performance gains tend to saturate or even slightly degrade on the GPQA dataset, indicating that $\alpha=1.0$ **provides a balance** between pattern guidance and general language modeling capabilities.
>
> We will include these results and analysis in the revised version to provide a comprehensive view of how $\alpha$ influences the reasoning process. Thank you again for your constructive suggestion.
>
> ## References
>
> [1] Ren, Yi, and Danica J. Sutherland. "Learning Dynamics of LLM Finetuning." The Thirteenth International Conference on Learning Representations.
>
> [2] Chen, Runjin, et al. "Seal: Steerable reasoning calibration of large language models for free." arXiv preprint arXiv:2504.07986 (2025).
>
> [3] Fu, Yichao, et al. "Efficiently Serving LLM Reasoning Programs with Certaindex." CoRR (2024).
>
> [4] Chen, Xingyu, et al. "Do NOT Think That Much for 2+ 3=? On the Overthinking of o1-Like LLMs." CoRR (2024).

---

> ### Author Response · Authors · 2025-11-28
> **Your feedback is appreciated**
>
> Dear Reviewer,
> Thank you again for your time and effort in reviewing our paper. We have clarified the questions raised and hope our rebuttal and the additional experiments have addressed your concerns. Please don’t hesitate to let us know if you have any further questions.

---

### Official Review · Reviewer_V4i9 · 2025-10-31

**Soundness:** 1
**Presentation:** 2
**Contribution:** 1
**Rating:** 2
**Confidence:** 4

**Summary:**

This paper introduces DynaGuide, a framework for efficiently distilling reasoning capabilities from large language models to smaller models through two main components: (1) Dynamic Data Selection (DDS), which adaptively selects valuable training samples during the first epoch of fine-tuning based on loss-driven weight adjustments across data feature classes, and (2) Reasoning Pattern Guidance (RPG), which addresses the overthinking problem in LLM-generated reasoning traces by introducing a projection loss that encourages execution steps over reflection and transition steps during fine-tuning. The authors demonstrate improvements across multiple model families (Qwen2.5, LLaMA-3.1, Qwen3) on knowledge-based question answering benchmarks using only 1,000 selected training examples from DeepSeek-R1 reasoning traces.

**Strengths:**

1. The paper addresses two important problems in reasoning capability distillation: accelerating the distillation process through data curation and mitigating the overthinking issue in synthetic reasoning traces generated by LLMs.
2. The experimental evaluation spans multiple model families (Qwen2.5, LLaMA-3.1, Qwen3) and different parameter scales, demonstrating that the approach is not limited to specific LLM architectures.
3. The paper includes ablation studies examining each component (DDS and RPG) separately, providing insights into their individual contributions to overall performance.

**Weaknesses:**

1. It is unclear why the paper proposes both DDS and RPG as a combined approach when they appear to be orthogonal components applied to different parts of the training pipeline. The relationship and necessity of using both together is not well justified.
2. Algorithm 1 mentions "FeatureClassOf" and the paper discusses data feature classes, but these concepts are not properly defined or explained elsewhere in the paper, making the implementation of the algorithm unclear.
3. In several places, the paper presents hypotheses to justify their methods without providing evidence or citations. For example, the cold start problem and issues with proper initialization in Section 3.1, weight explosion/vanishing in Section 3.2, and claims in Section 4.2 (after Equation 2) lack supporting evidence. Clear writing with motivating figures or citations for these hypotheses would strengthen the paper.
4. Experimental setups are not fully described. For example, in Table 1, it is unclear which DeepSeek-R1 model was prompted, with what prompt, on how many questions from the QA datasets, and how the percentages for execution/reflection/transition steps were calculated (LLM-as-a-judge, manual annotation, or rule-based analysis). Without fully specified experimental settings, the arguments in Section 4.1 motivating RPG are not completely sound.
5. Table 2 lacks important baseline comparisons. Each model (Qwen2.5-7B, LLaMA-3.1-8B, etc.) should show results when fine-tuned with only DDS and only RPG separately before showing the combined results. Additionally, any model distilled with 1,000 reasoning traces from the R1 model should demonstrate better performance on the evaluation sets in Table 2 due to in-domain fine-tuning, but these baseline experiments are not included (note: this is distinct from the s1 baselines).
6. The paper needs extensive ablation experiments on the number of training examples for DDS (currently fixed at 1,000) and the layer selection for applying RPG projection loss, which varies across different models. It is also unclear whether RAG evaluation is based on dense embedding search for documents or keyword lookup.

**Questions:**

1. What do the yellow lines represent in Figure 1? There is L_CLM shown in the blue box and L_total in the yellow box—are both losses being optimized simultaneously?
2. What is the effect of reasoning trace quality on the results? The paper only uses DeepSeek-R1 for generating reasoning traces.
3. What is "FeatureClassOf" in Algorithm 1? This function is not defined or explained in the paper. Why is it important to maintain weights over classes rather than individual data points?
4. Why was the specific functional form of f chosen in Equation (1)? Are there principled design choices or certain classes of functions that would work well with the paper's approach?
5. Line 233: How are the sentence parts encoded to extract hidden states?
6. Line 236: Wouldn't a 2D projection using t-SNE very weakly capture correlations in high-dimensional embedding spaces? Note that separation in 2D space does not necessarily imply separation in high-dimensional space.

### Typos and Editorial Suggestions
1. Lines 146-147: "Our fundamental premise ... few appearances" is unclear. A concrete example or plot would help clarify this statement.
2. Lines 157-161: The motivation regarding the cold start problem is not clear. A concrete example or plot would help demonstrate why the cold start is a problem.
3. Line 244: Typo in specifically

---

> ### Author Response · Authors · 2025-11-24
> **Author Response to Reviewer V4i9 (Part 1/4)**
>
> Thank you for your valuable feedback and constructive suggestions. We are committed to addressing these points to improve the quality of the paper.
>
> ## Response to Weakness 1
>
> We sincerely appreciate the reviewer's insightful question regarding the interplay between Dynamic Data Selection (DDS) and Reasoning Pattern Guidance (RPG). We argue that this combination is not arbitrary but **structurally necessary** for effective distillation.
>
> DDS adjusts the weights of data feature classes **through loss feedback**, aiming to expose the model to underperforming reasoning categories. However, **high loss can originate from not only valuable hard samples, but also harmful flawed synthetic data** (e.g., the "overthinking" in R1-generated chains). RPG functions as a robust learning objective that **mitigates the impact of these flawed data**. By incorporating specialized guidance, RPG enables the model to extract effective reasoning patterns even from the flawed data selected by DDS.
>
> **Our ablation studies strongly support this mechanism**. As shown in Table 4 of the paper, compared to DDS alone (Vanilla-SFT), the combined approach (DDS + RPG) achieves **higher accuracy** while **reducing the number of thinking tokens (average CoT length)**. This reduction directly proves that RPG successfully prunes the "overthinking" and redundant steps, and this brevity is due to more efficient reasoning, not information loss.
>
> ## Response to Weakness 2 & Question 3
>
> We thank the reviewer for pointing out these unclear points in the paper. We agree that clarifying these concerns promotes the understanding of our method.
>
> **1. Clarification on `FeatureClassOf` and Data Feature Classes**
>
> As mentioned in the first paragraph of Section 3, we characterize training samples along two orthogonal dimensions: **domain specificity** (e.g., Science, History, Law, Literature and Arts, etc.) and **task complexity** (The difficulty level of the question, determined either by dataset annotations, or proxied by the length of the CoT generated by the teacher model)
>
> `FeatureClassOf(d)` is the mapping function that maps a data point $d$
>  to its specific class based on these dimensions (e.g., FeatureClassOf(d) -> (History, Hard)). We will add these formal definitions to the paper and Algorithm 1 in the revised version.
>
> **2. Reason for maintaining weights over classes rather than individual data points**
>
> The reason why we choose to maintain weights at the class level rather than the instance level is related to the goal of Dynamic Data Selection (DDS). The core objective of DDS is to **select new, valuable data** from the candidate training data pool. Since we cannot calculate the loss for data points we haven't trained on yet, we cannot assign them individual weights. **By maintaining Class Weights**, we treat the average loss of seen samples in a class as a proxy for the **potential value of unseen samples in that same class**. If the model struggles with certain problems it has seen, selecting more samples with the same feature class from the pool might be beneficial for it.
>
> ## Response to Weakness 3
>
> We appreciate the reviewer’s rigorous attention to the theoretical and empirical foundations of our claims. We agree that substantiating these claims strengthens the paper. We address the specific points below with the corresponding evidence and citations.
>
> **1. The cold start problem and issues with proper initialization**
>
> The hypothesis that the model needs a stable initial phase before dynamic selection becomes effective is empirically **validated in our Ablation Study (Table 3)**.
>
> As shown in Table 3, **removing the "Warmup"** strategy leads to not only **a performance drop** but also **higher variance (instability)** in the results. This instability confirms that in the early stages of training, the loss signals are too noisy and fluctuate wildly. Without a warmup phase to establish a stable representation, the dynamic selection algorithm overfits to this initial noise, leading to erratic optimization trajectories.
>
> **2. Weight Explosion/Vanishing**
>
> **This is a fundamental property of multiplicative weight updates.** In our algorithm, feature class weights are updated via multiplication ($w_{i+1} \leftarrow w_i \cdot f_i$). If a data point exhibits extreme loss values (which is common in hard reasoning tasks during training), the multiplier will deviate from 1, leading to exponential growth (explosion) or decay toward zero (vanishing). We introduce the sublinear weight adjustment factor function specifically to counteract this numerical instability.
>
> **Due to character limitations, our response continues in the following comment.**

---

> ### Author Response · Authors · 2025-11-24
> **Author Response to Reviewer V4i9 (Part 2/4)**
>
> Continuing from the previous comment, here we provide our responses to the rest questions.
>
> **3. Claims in Section 4.2**
>
> We claim that the projection loss encourages the model to generate more execution steps, which is **grounded in the field of Representation Engineering (RepE) or activation steering** [1][2]. These works demonstrate that by identifying activation patterns associated with specific behaviors and fine-tuning the model to align with them, one can steer model outputs. We apply this principle: RPG finetunes the hidden states that act as steering vectors to promote execution steps and suppress the "overthinking".
>
> We will incorporate these explanations and references into the revised version to strengthen the foundation of our work and enhance its clarity.
>
> ## Response to Weakness 4
>
> We would like to clarify that the data used for the analysis in Table 1 (Section 4.1) is **identical with the training data** described in Section 5.1.
>
> **1. Which DeepSeek-R1 model was prompted**
>
> We use the DeepSeek-R1-671B model (official release dated January 20, 2025) as the teacher model for generating reasoning chains. We will add the specific DeepSeek-R1 version to Experimental Setup in the revision.
>
> **2. With what prompt**
>
> The exact prompt used to query DeepSeek-R1 is listed in Appendix C.3 of the submitted paper.
>
> **3. On how many questions from the QA datasets**
>
> We conduct this analysis on the full training set. As detailed in the first paragraph of Section 5.1, it includes 71,662 samples from StrategyQA, HotpotQA, and SuperGPQA.
>
> **4. How the percentages for execution/reflection/transition steps were calculated**
>
> As discussed in Section 4, we employ a rule-based keyword matching approach to categorize steps, following the methodology established by *Chen et al.* [3] (cited in the first paragraph of Section 4).
>
> ## Response to Weakness 5
> We thank the reviewer for the constructive suggestions regarding the presentation of our baselines. We agree that showing the performance of individual modules before that of the entire framework provides a clearer view.
>
> We would like to clarify that, to demonstrate the separate effects of DDS and RPG, the **detailed analysis is included in ablation study (Section 6)** on Qwen2.5-7B-Instruct as the representative backbone due to space constraints.
>
> Regarding the **In-Domain Fine-Tuning Baseline**, we clarify that it corresponds exactly to the "Random" method shown in Table 3, which represents Vanilla SFT on 1,000 randomly sampled reasoning traces, without any specific selection strategy or loss guidance. This serves as the direct standard comparison for our method.
>
> ## Response to Weakness 6
> We thank the reviewer for the detailed questions regarding our experimental settings. We address these three points below to clarify our methodology.
>
> **1. Rationale for Training Size**
>
> We acknowledge the reviewer's interest in scaling effects. We fixed the training size at 1,000 samples to **align with the s1 [4] baseline setting**. This constraint ensures a controlled, fair comparison and highlights the data efficiency of our framework.
>
> **2. Layer selection for RPG**
>
> Regarding the selection of layers for the RPG projection loss, we clarify that these are not arbitrary hyperparameters requiring random search or ablation.
>
> As described in Section 4.2 and Appendix E, the specific layer for each backbone model is determined through a probe analysis on the linear separability of representations from shallow layers to deep layers.
>
> **3. RAG evaluation**
>
> We clarify that our RAG evaluation setup utilizes **gold context** provided by the dataset metadata, rather than an external retrieval system (dense or keyword-based). We crawled the text content from the **reference URLs** provided in the dataset. This content is truncated to fit an 8K token window and directly prepended to the question as context to evaluate the model's long-context reasoning ability.
>
> ## Response to Question 1
>
> The yellow dashed box in Figure 1 illustrates the Reasoning Pattern Guidance (RPG) training phase. **The yellow lines (arrows) represent the flow of data tensors and gradient computation**: (1) Data flows from the Training Dataset into the model; (2) The model computes the standard causal language modeling loss ($\mathcal{L}\_{\text{CLM}}$) and the hidden states of tokens containing '\\n\\n' ($H_c$); (3) $H_c$ are compared with pre-calculated execution-step feature vector $H_E$ to compute the projection loss ($\mathcal{L}\_{\text{proj}}$); (4) These terms are aggregated into the total loss ($\mathcal{L}\_{\text{total}}=\mathcal{L}\_{\text{CLM}} + \alpha \cdot \mathcal{L}\_{\text{proj}}$), which is used to update the model parameters via back propagation.
>
> The blue box (DDS) and the yellow box (RPG) represent two distinct stages within the framework, and they are not optimized simultaneously.
>
> **Due to character limitations, our response continues in the following comment.**

---

> ### Author Response · Authors · 2025-11-24
> **Author Response to Reviewer V4i9 (Part 3/4)**
>
> Continuing from the previous comment, here we provide our responses to the rest questions.
>
> ## Response to Question 2
>
> **1. Effect of Reasoning Trace Quality**
>
> **We have already investigated the impact of reasoning trace quality in Section 6.3 (Table 5)** of the submitted paper. We conducted an ablation study comparing distillation performance when training on Correct Reasoning Traces versus Incorrect Reasoning Traces (traces that follow a reasoning structure but lead to a wrong answer). As shown in Table 5, the model distilled from incorrect traces still maintains competitive performance. This result suggests that our framework works even on imperfect reasoning traces, and it enables the model to learn the general reasoning capability, rather than merely memorizing factual knowledge or gold answers.
>
> **2. The paper only uses DeepSeek-R1**
>
> We acknowledge that conducting experiments on other teacher models would be valuable. However, **as discussed in Section 7, our choice was constrained by the availability** of full, transparent Chain-of-Thought (CoT) data. Currently, DeepSeek-R1 is the state-of-the-art open-weights model that provides access to its **complete raw reasoning traces**. Other top-tier reasoning models (e.g., OpenAI o1, o3) are closed-source and typically hide their internal reasoning chains or only provide a summarized version. Since our method (and the baseline s1) relies on distilling the detailed step-by-step CoT, DeepSeek-R1 was the only viable high-performance candidate at the time of experimentation.
>
> ## Response to Question 3
> Please see the above "Response to Weakness 2 & Question 3".
>
> ## Response to Question 4
> As discussed in Section 3.2, the primary constraint for designing $f$ is that it should be **sublinear with respect to the loss ratio to mitigate weight explosion or weight disappearance**. Therefore, any function that is monotonic, sublinear, and bounded would theoretically work well with our approach. Our specific choice in Equation (1) is a representative instance of this class that empirically balances responsiveness with stability.
>
> ## Response to Question 5
> To extract the hidden states mentioned in Line 233, we input the entire sequence (the full reasoning trace) into the model and extract the hidden states of tokens containing '\n\n' at all the layers.
>
> Specifically: (1) We input the complete token sequence $T=[t_1,t_2,\dots,t_n]$ into the model. This yields the hidden states for every token at every layer: $H=\\\{h_i^l \mid 1 \le i \le n, 1 \le l \le L\\\}$, where $L$ is the total number of layers in the model. (2) We identify the indices $J=\\\{j_1,j_2,\dots,j_m\\\}$ corresponding to the step delimiter tokens (tokens containing '\n\n'), which mark the boundaries of reasoning steps. (3) We then extract the hidden states at these specific indices $H_{\text{extracted}}=\left\\{\left[h_{j_1}^{l},h_{j_2}^{l},\dots,h_{j_m}^{l}\right]\mid 1 \le l \le L\right\\}$.
>
> These extracted vectors are then used for the subsequent analysis described in Section 4.2, such as linear separability probing, to identify the optimal layer for the RPG loss.
>
> ## Response to Question 6
> We appreciate the reviewer’s caution regarding the interpretation of dimensionality reduction plots. While we agree that 2D projections are approximations, we suggest that **the separability observed in 2D projection serves as a lower bound for the separability in the original high-dimensional space**.
>
> Our argument relies on **the geometric properties of high-dimensional vector spaces**. High-dimensional spaces are inherently sparse. Two randomly selected unit vectors in a high-dimensional space are highly likely to be nearly orthogonal (with their inner product $\to 0$ ). Consequently, **data points that appear separable in a constrained 2D projection are almost more separable in the original high-dimensional space due to the information compression of the projection**.
>
> ## Response to Typos and Editorial Suggestions
>
> We thank the reviewer for pointing out these areas where clarity could be improved. We address them below.
>
> 1. We mean that **a simple pattern can be learned quickly** by the model. Its loss drops rapidly early in training, while **a complex transition can make the model struggle**,  resulting in a loss that decreases much more slowly.
>
> 2. The cold start problem arises because our DDS module depends on historical loss statistics to assign weights, but there is no history at the very beginning of training. Without a warmup phase, the first few batches determine the weight distribution based on randomness rather than true data difficulty. If the first batch happens to be hard, the weights might spike extremely, destabilizing the training. Our ablation study (Table 3) shows the necessity of the warmup phase.
>
> 3. Thank you for pointing out the typo. We will correct the spelling of "specifically" in the revised version.
>
> **Due to character limitations, our response continues in the following comment.**

---

> ### Author Response · Authors · 2025-11-24
> **Author Response to Reviewer V4i9 (Part 4/4)**
>
> Continuing from the previous comment, here we provide the references.
>
> ## References
>
> [1] Zou, Andy, et al. "Representation engineering: A top-down approach to ai transparency." arXiv preprint arXiv:2310.01405 (2023).
>
> [2] Bartoszcze, Lukasz, et al. "Representation Engineering for Large-Language Models: Survey and Research Challenges." arXiv preprint arXiv:2502.17601 (2025).
>
> [3] Chen, Runjin, et al. "Seal: Steerable reasoning calibration of large language models for free." arXiv preprint arXiv:2504.07986 (2025).
>
> [4] Muennighoff, Niklas, et al. "s1: Simple test-time scaling." Proceedings of the 2025 Conference on Empirical Methods in Natural Language Processing. 2025.

---

> ### Author Response · Authors · 2025-11-28
> **Your feedback is appreciated**
>
> Dear Reviewer,
> Thank you again for your time and effort in reviewing our paper. We have clarified the questions raised and hope our rebuttal have addressed your concerns. Please don’t hesitate to let us know if you have any further questions.

---

### Official Review · Reviewer_JcUx · 2025-10-31

**Soundness:** 3
**Presentation:** 3
**Contribution:** 3
**Rating:** 8
**Confidence:** 2

**Summary:**

The paper addresses the problem of increasing computational demands of LLM distillation, such as expensive training, annotation, suboptimal data selection, etc. The paper introduces DynaGuide which optimizes distillation in terms of efficiency and performance. The method integrates dynamic data selection during training and reasoning pattern guidance that mitigates overthinking in synthetic data. Together these methods showcase efficient data distillation with a small set of selected samples.

**Strengths:**

- The paper addresses a crucial problem of LLM distillation efficiency
- The paper proposes a simple and elegant solution of dynamic data selection and reasoning pattern guidance
- The paper is well written and clear, it is easy to follow

**Weaknesses:**

The evaluation is limited to QA datasets.

**Questions:**

Does the proposed framework also show strong empirical performance on datasets beyond the QA datasets?

---

> ### Author Response · Authors · 2025-11-24
> **Author Response to Reviewer JcUx**
>
> We thank the reviewer for acknowledging the strengths of our work and suggesting an **Accept** rating. We are also grateful for your insightful question about the generalization of DynaGuide beyond standard QA tasks.
>
> We would like to clarify that although our benchmarks are formatted as "Question-Answering," **they cover a diverse range of underlying reasoning capabilities**, including logical deduction, scientific reasoning, and mathematical problem-solving, rather than simple knowledge retrieval.
>
> **The experiments in our paper have included results on the GPQA dataset**, a rigorous benchmark for expert-level scientific and mathematical reasoning covering physics, biology, and chemistry, which are significantly distinct from traditional open-domain QA. As shown in Table 2 of the paper, DynaGuide achieves significant gains on GPQA, demonstrating that **our framework enhances complex reasoning patterns**, not just a QA-specific method.
>
> To fully address your concern, we have conducted **additional experiments on the AIME-2024 benchmark**, which focuses strictly on multi-step mathematical reasoning. As shown in the table below, DynaGuide achieves superior performance compared to standard baselines. Given the challenging nature of AIME 2024, these consistent improvements across multiple backbones confirm the robustness of our method.
>
> **Table: Performance comparison on AIME 2024.**
> |Model|Fine-Tuning Data| AIME 2024 Performance |
> |:--:|:--:|:--:|
> ||***Qwen2.5-7B-Instruct Series***|
> | Qwen2.5-7B-Instruct | - | 16.7 |
> | s1-Qwen2.5-7B + BF | 1,000 | 23.3 |
> | **DynaGuide-Qwen2.5-7B (ours)** | 1,000 | 30.0 |
> ||***LLaMA-3.1-8B-Instruct Series***|
> | LLaMA-3.1-8B-Instruct | - | 3.3 |
> | s1-LLaMA-3.1-8B + BF | 1,000 | 6.7 |
> | **DynaGuide-LLaMA-3.1-8B (ours)** | 1,000 | 16.7 |
> ||***Qwen3-4B Series***|
> | Qwen3-4B | - | 60.0 |
> | s1-Qwen3-4B + BF | 1,000 | 66.7 |
> | **DynaGuide-Qwen3-4B (ours)** | 1,000 | 76.7|
> || ***Qwen3-8B Series*** |
> | Qwen3-8B | - | 73.3 |
> | s1-Qwen3-8B + BF | 1,000 | 76.7 |
> | **DynaGuide-Qwen3-8B (ours)** | 1,000 | 80.0 |
>
> DynaGuide demonstrates strong empirical performance on diverse reasoning tasks, including mathematics (AIME 2024) and rigorous science (GPQA), verifying its effectiveness beyond traditional QA datasets.
>
> We hope that these clarifications effectively address your concerns. We stand ready to answer any follow-up questions immediately.

---

### Author Response · Authors · 2025-11-28
**Kind Reminder Regarding Our Rebuttal**

Dear Reviewers,

Thank you very much for your thoughtful and constructive comments on our submission. We have carefully addressed each point raised and provided detailed clarifications and updates in our rebuttal.

As the discussion deadline is approaching, we kindly ask whether you could take a moment to review our responses at your convenience. We would greatly appreciate any follow-up feedback regarding whether our clarifications sufficiently resolve your concerns.

Thank you again for your time and effort in reviewing our work. We sincerely value your input.

Best regards,

The Authors

---

### Meta-Review · Area_Chair_ZBHr · 2026-01-06

**Summary:**

The paper proposes improvements for reasoning LLM’s distillation into more compact models based on dynamic data selection and guidance of reasoning patterns to avoid overthinking. The reviewers appreciated practical improvements coming from the proposed approaches, and raised concerns around connection between the two proposed mitigations and substantial support of claims and hypotheses.

**Reviewer Concerns:**

See below.

**Reviewer Scores:**

Reviewer JcUx provided an extremely short and non detailed review but raised the concern that paper’s evaluation was limited to QA datasets (which is a somewhat unclear concern, since they did not specify what other evaluation they were interested in). The authors response adequately addressed this question by providing additional AIME results and highlighting GPQA results. I believe the reviewer would retain score 8, however, I consider this review with low weight since it is not detailed.

The reviewer V4i9 raised several concerns: 1) the motivation behind combining two independent tricks for distillation into one paper, 2) unsupported claims / hypotheses, 3) stronger baselines and ablation experiments. I believe the authors’ rebuttal partially addressed 2) and 3) but didn’t provide a convincing response to 1). I believe the reviewer would increase 2->4.

Reviewer ywwV raised concerns related to clarity and support of claims. The authors reasonably addressed most questions and I believe the reviewer would raise the score 2->4.

The reviewer 9ndd raised questions about method details, and experimental setup and baselines. I believe the authors responded to the questions well and the reviewer would retain a score of 6.

I believe given outstanding concerns the paper is borderline and I recommend a rejection given lack of motivation for proposed combination of approached and support of claims.

---

### Decision · Program_Chairs · 2026-01-26

Reject